# Effect of Electroejaculation Protocols on Semen Quality and Concentrations of Testosterone, Cortisol, Malondialdehyde, and Creatine Kinase in Captive Bengal Tigers

**DOI:** 10.3390/ani13121893

**Published:** 2023-06-06

**Authors:** Jaruwan Khonmee, Janine L. Brown, Anabel López Pérez, Teepakorn Lertwichaikul, Anucha Sathanawongs, Patchara Pornnimitra, Chanakan Areewong, Jarawee Supanta, Veerasak Punyapornwithaya, Songphon Buddhasiri, Khanittha Punturee

**Affiliations:** 1Department of Veterinary Bioscience and Veterinary Public Health, Faculty of Veterinary Medicine, Chiang Mai University, Chiang Mai 50100, Thailand; 2Center of Elephant and Wildlife Health, Chiang Mai University-Animal Hospital, Chiang Mai 50100, Thailand; 3Elephant, Wildlife, and Companion Animals Research Group, Chiang Mai University, Chiang Mai 50100, Thailand; 4Center for Species Survival, Smithsonian National Zoo and Conservation Biology Institute, Front Royal, VA 22630, USA; 5Elephant Conservation Center, Sayaboury 08000, Laos; 6Tiger Kingdom Chiang Mai, Khum Suea Trakan Co., Ltd., Chiang Mai 50180, Thailand; 7Department of Medical Technology, Faculty of Associated Medical Sciences, Chiang Mai University, Chiang Mai 50200, Thailand

**Keywords:** *Panthera tigris*, tigers, Bengal tiger, electroejaculation, semen, testosterone, cortisol, malondialdehyde

## Abstract

**Simple Summary:**

The Bengal tiger (*Panthera tigris tigris*) is critically endangered, so artificial insemination (AI) is an important conservation tool. Electroejaculation (EE) protocols have been optimized to obtain sufficient amounts of viable sperm for artificial insemination in many felid species; however, less attention has been paid to animal wellbeing during the process. This study examined the effects of three EE protocols (Low, 2–5 volts; Medium, 3–6 volts; and High, 4–7 volts) on semen quality, testicular size, serum creatine kinase (CK, as a proxy for muscle damage), serum testosterone, and cortisol (as a proxy for stress) concentrations, and seminal plasma and sperm pellet malondialdehyde (MDA, as a measure of oxidative stress) measured after each EE series. Primary sperm abnormalities and seminal plasma MDA were higher in the Low compared to Medium and High voltage groups (*p* < 0.05). Serum CK in the High voltage group increased during the EE series suggesting the potential for muscle damage (*p* < 0.05). However, no significant differences were observed for serum cortisol, testosterone, or MDA concentrations across voltage groups. Results suggest the Medium voltage protocol produced good quality samples at lower voltages and with no increase in CK compared to the High protocol, which might be better for animal welfare.

**Abstract:**

The Bengal tiger (*Panthera tigris tigris*) is critically endangered, so assisted reproductive technologies, including artificial insemination, are important conservation tools. For wild and domestic felids, electroejaculation (EE) is the most common semen collection method, with protocols optimized to obtain sufficient amounts of viable sperm for artificial insemination. However, less attention has been paid to ensuring animal wellbeing during the process. This study examined the effects of three EE protocols (Low, 2–5 volts; Medium, 3–6 volts; High, 4–7 volts) on semen quality, testicular size, serum testosterone, creatine kinase (CK), and malondialdehyde (MDA) concentrations, and serum cortisol as a proxy for stress. Blood samples were collected before, during, and after each EE series. Seminal plasma pH, and sperm motility, viability, and morphology were evaluated after each procedure. Seminal plasma and sperm pellet MDA concentrations were also determined. Primary sperm abnormalities and seminal plasma MDA were higher in the Low compared to Medium and High voltage groups (*p* < 0.05). Serum CK in the High voltage group increased during the EE series (*p* < 0.05), suggesting the potential for muscle damage. However, no significant changes were observed for serum cortisol, testosterone, or MDA concentrations. Results suggest the Medium voltage protocol produced good quality samples at lower voltages than the High protocol with no negative effect on muscle function, which might be better for animal welfare.

## 1. Introduction

The tiger (*Panthera tigris*) is classified as endangered [1] and one of the most threatened large carnivores in the world due to habitat loss, decreasing prey availability [2,3,4], and poaching [5]. Tiger population declines and fragmentation into smaller, isolated groups significantly hinder gene flow and, thus, threaten the species’ biodiversity [6]. Since 2004, Thailand has increased its tiger conservation efforts and undertaken more rigorous enforcement, monitoring, and research for their protection [7]. However, only about 190–250 animals are believed to exist in situ in Thailand, according to Pisdamkham et al. [7]. By contrast, upwards of 2000 Bengal tigers are held in zoos and other tourist venues throughout the country [8]. While unlikely to be considered for reintroduction, ex situ populations still offer important research opportunities to drive conservation strategies for the protection and survival of wild counterparts and other tiger subspecies.

The development of assisted reproductive technologies for wildlife species, including artificial insemination (AI), has been proposed as a way to overcome challenges in managing small, isolated populations that reproduce poorly [9,10] and ensure genetically valuable individuals pass genes on to the next generation [11,12]. However, the success of AI remains inconsistent for many wild Felidae, including tigers [11], in part because of variable sperm quality in response to semen collection techniques [13,14]. Therefore, more studies are needed to improve semen collection protocols to provide adequate samples for successful AI in this species [15].

For wild and domestic felids, electroejaculation (EE) is the most common semen collection technique [14,16,17,18]. Initially developed at the Smithsonian’s National Zoological Park [13,19], it has been used successfully in several felid species, including tigers [15,20,21,22,23]. Although EE is an effective technique, it can potentially cause pain and stress to the males being collected [24,25]. In other species, reactions to voltages used with EE include changes in heart, pulse, and respiratory rates; oxygen saturation; rectal temperature; alkaline phosphatase; and cortisol concentrations [26]. In addition, creatine kinase (CK), an enzyme found in muscles that increases after a heart attack, skeletal muscle injury, or strenuous exercise [26,27], could be related to intense muscle contractions noted with EE procedures. Therefore, concurrent monitoring of stress and reproductive biomarkers can help to assess physical and physiological responses, and thus wellbeing, to various EE protocols used with wildlife as part of species conservation programs.

Glucocorticoids (GCs) are steroids produced by the adrenal cortex that affect nearly every organ and tissue in the body, regulating diverse physiologic processes such as energy metabolism, immune responses, reproduction, behavior, cell proliferation, and cardiovascular function [28]. They could also provide information on physiological stress responses during EE. Hyperadrenal activity related to chronic stress can adversely influence male reproductive performance through disruptions of the hypothalamic–pituitary–gonadal axis [29,30,31], particularly testosterone, an essential hormone for maintaining spermatogenesis and male fertility [32]. In short-term evaluations, Wildt et al. [14] showed variable effects of EE on testosterone concentrations across felid species, being insignificant in cheetahs (*Acinonyx jubatus*) and pumas (*Puma concolor*), suppressive in leopards (*Panthera pardus*), and inconsistent in tigers. However, little is known about how varying voltage EE protocols affect physiological function, particularly that related to the adrenal–testicular system.

Lipid peroxidation has been proposed to negatively affect sperm quality because the plasma membrane of mammalian sperm contains abundant polyunsaturated fatty acids that are susceptible to attack by free radicals [33,34,35]. Malondialdehyde (MDA) is a product of lipid peroxidation, with elevated concentrations in seminal plasma being associated with reduced semen quality in humans [36,37], dogs [38], rams [39], and Asian elephants [40]. To our knowledge, relationships between semen quality and MDA have not yet been examined in tigers.

This study aimed to develop a safe, reliable method of semen collection in the tiger by comparing ejaculate traits in response to three EE protocols of varying voltages. A second aim was to measure serum cortisol, testosterone, MDA, seminal plasma MDA, and serum CK before and after each EE series to determine the effects on adrenal–testicular function and muscular function. The ultimate goal is to identify an effective EE protocol for collecting good-quality semen samples from tigers that has minimal adverse effects on physiological function and welfare.

## 2. Materials and Methods

### 2.1. Ethics Statement

This study was approved by the Faculty of Veterinary Medicine, Chiang Mai University Animal Care and Use Committee (FVM-ACUC; permit number R12/2560). Permission to conduct the study on tigers at the Tiger Kingdom Chiang Mai was obtained from the veterinary staff and mammal curator, who were collaborators on the project.

### 2.2. Animals

Twelve healthy male Bengal tigers (7.0 ± 0.9 years of age; range, 4.3–7.0 years) were used in this study. All were retired animals, no longer involved in tourist activities. None of the males had ever bred or been part of semen collection procedures before. The tigers were wormed every 3 months and vaccinated against feline herpesvirus 1, feline calicivirus, and feline panleukopenia virus. They also received annual health exams conducted by veterinarians at the facility.

All tigers were maintained in single, off-exhibit enclosures, 4 m × 4 m × 4 m in size with a 20 m × 5 m outside yard. Males were housed adjacent to females in the same complex but were separated by wire mesh. Therefore, all tigers had visual, olfactory, and auditory interactions with conspecifics. The enclosures had a concrete base and at least one climbing log. Animals were exposed to natural light and fed a diet of pork meat and chicken carcass (3 kg) mixed with carnivore premix daily. Water was provided ad libitum. As a daily routine, each tiger was allowed to play in a grass field and swim in a large pool for 30 min to 1 h per day, without contact with other tigers. The rest of the day, the tigers were kept inside their enclosures with various forms of enrichment, such as wooden platforms, dummy prey to stimulate hunting behavior, and hiding food throughout the enclosure.

### 2.3. Anesthesia for Semen Collection

Tigers were anesthetized using zolazepam–tiletamine (Zoletil^®^, Virbac, France) 0.7–1.0 mg/kg combined with xylazine HCl (RompunTM, Bayer, Germany) 0.7–1.0 mg/kg delivered by a blow dart in off-exhibit enclosures. Acetate Ringer’s solution as a fluid supplement was administered intravenously with a 22-gauge IV catheter. Anesthesia was maintained with a combination of zolazepam–tiletamine and ketamine 5–10 mg/kg i.v. (Troy labs, Australia). Vital signs, including heart rate, respiratory rate, capillary refilling time, and temperature, were monitored every 5 min during the anesthesia period until the animals gained consciousness. Once semen collection was completed (~20–25 min), anesthesia was reversed by yohimbine (Sigma-Aldrich, Milano, Italy) via i.m. injection in equal amounts to xylazine.

### 2.4. Testicular Size Measurement

The size of each testis was measured through the scrotal wall before each collection by measuring length, width, and height using a standard Vernier caliper. Testicular volume was calculated using the Lambert equation (length [L] × width [W] × height [H] × 0.71).

### 2.5. Electroejaculation Protocols

Three EE protocols of varying voltages were used based on previously described regimens [16] (n = 12 tigers/group). Each animal received 80 electrical stimuli delivered by a 60 HZ sinewave electroejaculator (Minitube^®^, Tiefenbach, Germany) with a rectal probe 55 cm in length and 3.7 mm wide inserted approximately 22 cm into the rectum. The 80 stimuli were administered in a 2-s-on/2-s-off pattern and divided into three series of 30, 30, and 20 stimuli of increasing voltages, with a 3 min rest period between series. Tigers were randomly selected to receive a Low (Series 1: 2, 3, 4 V; Series 2: 3, 4, 5 V; Series 3: 4, 5 V), Medium (Series 1: 3, 4, 5 V; Series 2: 4, 5, 6 V; Series 3: 5, 6 V), or High (Series 1: 4, 5, 6 V; Series 2: 5, 6, 7 V; Series 3: 6, 7 V) protocol. All tigers were exposed to each EE protocol with a 1-month break from June to September. The tigers were rotated for each EE protocol belonging to the experiment period.

### 2.6. Semen Evaluation

Semen collected after each series was pooled; therefore, only one sample per tiger per EE protocol was evaluated. Pooled samples were immediately assessed for color, volume, pH, sperm concentration, total spermatozoa/ejaculate, sperm motility, sperm viability, and the percentage of normal and abnormal spermatozoa. The same person conducted all semen evaluations.

Ejaculate volume was measured using a calibrated positive displacement pipette, pH was determined using pH paper, and sperm concentration was assessed using a SpermaCue photometer (Minitube^®^, Tiefenbach, Germany). Sperm motility was evaluated in a 20 µL semen drop on a prewarmed slide (37 °C) with a coverslip using phase-contrast microscopy (Olympus BX 51, Tokyo, Japan) at 100× magnification [41].

Sperm viability was assessed by diluting semen with an equal volume of eosin–nigrosin stain (1% eosin, 3% nigrosin, 3% sodium citrate, and 100 mL distilled water) using an applicator stick. Lastly, a thin uniform smear was made followed by air-drying. A total of 200 spermatozoa were evaluated for live (unstained) and dead (stained) sperm heads under bright field microscopy at 400× magnification [42].

The percentage of sperm abnormalities was evaluated in a total of 200 spermatozoa per sample and was determined using a phase contrast microscopy at 400× and 1000× magnification. Abnormalities were divided into primary (detached heads, excessively large or small heads, and misshapen heads) and secondary (proximal or distal cytoplasmic droplets, bent tails, coiled tails, and bent midpieces) forms according to the World Health Organization classification system [43].

### 2.7. Blood Collection

Blood samples (8 mL) were collected from the femoral vein by trained veterinarians at the Tiger Kingdom or Chiang Mai University. Samples were collected while the tiger was sedated during the semen collection procedure. A total of five blood samples were collected per individual: one within 10 min before of the first EE series, one after each EE series, and one 10 min after the EE procedure was completed. Blood samples were kept at 4 °C and centrifuged at 3000× *g* for 10 min within a few hours of collection. The serum was stored at −20 °C until further analysis.

### 2.8. Cortisol Analyses

Serum cortisol was measured using a double-antibody EIA with a polyclonal rabbit anticortisol antibody (R4866, Coralie Munro, University of California). Second antibody-coated plates were prepared by adding 150 μL of antirabbit IgG (0.01 mg/mL) to each well of a 96-well microliter plate, and incubating at room temperature (RT) for 15–24 h. The wells were then emptied and blotted dry, adding 250 μL blocking solution and incubating for 15–24 h at RT. After incubation, all wells were emptied, blotted, and dried in a dry cabinet at RT (Sanpla Dry Keeper, Sanplatec Corp., Auto A-3, Japan) with loose desiccant in the bottom. After drying (humidity < 20%), the plates were heat-sealed in a foil bag with a 1 g desiccant packet and stored at 4 °C until use.

Serum samples and cortisol standards (50 μL) were added to the appropriate wells. cortisol–horseradish peroxidase (HRP, 25 μL, 1:16,000) was added immediately to each well, followed by 25 μL polyclonal rabbit anticortisol antibody (R4866) to all but nonspecific binding wells, and incubated on a shaker (SHO-2D, Gangwon-do, Republic of Korea) at RT for 1 h. The plates were washed five times with wash buffer (1:20 dilution, 20X Wash Buffer Part No. X007; Arbor Assays, MI, United States) followed by the addition of 100 μL TMB substrate solution and incubation for 20–30 min at RT without shaking. After that, 50 μL of stop solution (2M H_2_SO_4_) was added to all wells. The absorbance was measured at 450 nm with a microplate reader (TECAN, Grödig, Austria). The assay was validated for tiger serum by demonstrating parallelism between serial dilutions of serum and the cortisol standard curve based on Pearson’s correlation coefficient analyses (r > 0.95). Adding unlabeled cortisol standards to pooled tiger serum resulted in a significant recovery of mass (>90%). Assay sensitivity (based on 90% binding) was 0.43 ng/mL. Samples were analyzed in duplicate; intra- and interassay CVs were <10% and <15%, respectively.

### 2.9. Testosterone Analyses

Serum testosterone concentrations were quantified by a double-antibody EIA utilizing a polyclonal rabbit antitestosterone antibody (R156/7, C. Munro, 25 μL, 1:110,000 dilution) and testosterone–horseradish peroxidase (HRP, 25 μL, 1:20,000 dilution) label. Serum samples were diluted 1:1 with assay buffer for analysis in duplicate (50 μL), and absorbance was measured at 450 nm. The assay was validated for tiger serum by demonstrating parallelism between serial dilutions of serum and the testosterone standard curve based on Pearson’s correlation coefficient analyses (r > 0.95). The addition of unlabeled testosterone standards to pooled serum resulted in a significant recovery of mass (>90%). Assay sensitivity (based on 90% binding) was 0.12 ng/mL, and the intra- and interassay CVs were <10% and <15%, respectively.

### 2.10. Malondialdehyde Analyses

MDA in neat serum and seminal plasma and 500 µL of PBS with sperm pellets were quantified by a thiobarbituric acid reacting substances (TBARS) test described by Satitmanwiwat et al. [44]. Briefly, 50 µL of samples and standards were mixed with 750 µL of 0.44 M phosphoric acid, 250 µL of 42 mM thiobarbituric acid (TBA), and 450 µL of distilled water. The samples were boiled for 15 min, cooled on ice for 5 min, and centrifuged at 1500 rpm for 5 min. The supernatant was collected and measured at 532 nm using a UV-VIS spectrophotometer (Shimadzu, Kyoto, Japan). MDA concentrations were calculated from standard curves of MDA equivalents generated by the acid-catalyzed hydrolysis of 1,1,3,3-tetramethoxypropane (TMP) (5–80 µM).

### 2.11. Muscle Enzymes Analyses

Serum creatine kinase (CK) was measured in an Automated Clinical Chemistry Analyzer (Sysmex; BX-3010, Bangkok, Thailand).

### 2.12. Statistical Analyses

Data are reported as the mean ± standard error of the mean (SEM). Statistical analyses were performed using R version 4.0.3 [45]. Repeated measures data were analyzed using Generalized Least Squares (GLS) [46] to determine the effects of EE protocols on semen quantity and quality, and testosterone, cortisol, MDA, and CK concentrations. The GLS model included EE protocols and time as fixed effects, whereas the individual animal was defined as a random effect. The autoregressive of order 1 (AR1) correlation structure was defined in the GLS model as it provided the lowest values of Akaike Information Criterions (AIC). Mean differences were further analyzed using Tukey’s post-hoc tests. In addition, correlations among testosterone and cortisol concentrations, testosterone and MDA concentrations, and cortisol and MDA concentrations calculated as aggregated data were analyzed using Pearson’s correlation tests. Statistical significance was set at *p* < 0.05.

## 3. Results

### 3.1. Semen Quality and Testicular Size

A total of 30 semen samples were collected among the three voltage protocols (Low = 9, Medium = 9, High = 12). Primary sperm abnormalities and seminal plasma MDA concentrations were higher in the Low voltage group than in the Medium and High voltage groups (*p* < 0.05). Other than those parameters, no other differences were found for ejaculation volume, sperm concentration or total number, motility, viability, pH, secondary abnormalities, MDA concentrations in sperm pellets (Table 1), or testicular size (Table 2) across the three EE protocols. The majority of primary sperm abnormalities in pooled samples were macrocephalic heads and coiled tails.

### 3.2. Serum Testosterone, Cortisol, MDA, and CK Concentrations

A total of 178 serum samples were collected among the three voltage protocols. Serum testosterone concentrations were similar before, during, and after each collection series across the three EE protocols. However, overall mean testosterone concentration was lower in the Medium voltage group compared with the High and Low voltage series protocols. Mean CK concentration in the High voltage group was significantly higher at the end of the EE series compared to the beginning (*p* < 0.05) (Table 3). Neither serum cortisol nor MDA concentrations were affected by any of the EE protocols or voltage series (*p* > 0.05) (Table 3). Finally, based on Pearson’s correlation tests, there were no significant correlations among the serum biomarker measures.

### 3.3. MDA Concentrations and Semen Parameters

There were no relationships among MDA in serum, seminal plasma, or sperm pellets and seminal traits, although several approached significances: MDA and secondary abnormalities in serum (r = 0.76, *p* = 0.07) and sperm pellets (r = −0.87, *p* = 0.05), albeit in opposite directions; MDA and percent normal (r = −0.79, *p* = 0.06); and MDA and sperm concentration (r = 0.61, *p* = 0.07).

## 4. Discussion

This study investigated the effects of three EE protocols (Low: 2–5 V; Medium: 3–6 V; and High: 4–7 V) on semen characteristics (volume, concentration, motility, viability, primary, and secondary abnormalities), serum CK, testosterone, cortisol, and MDA concentrations, and MDA in sperm pellets and seminal plasma in male Bengal tigers. Overall, seminal traits were within the range of previously published studies utilizing comparable collection techniques in tigers [14,15], with the exception of seminal volume (lower) and pH (higher). Based on biochemical tests, there were no significant differences in serum cortisol or MDA concentrations before, during, or after each EE series or between voltage protocols; however, testosterone was lower in the Medium compared to the High voltage group. By contrast, serum CK concentrations increased during the EE series for the High voltage group, suggesting negative effects on muscle function and possible damage. Overall, the Medium voltage protocol appeared to be the most effective for producing good quality samples at a lower voltage with no negative effect on muscle function, which could potentially be better for animal welfare.

Ejaculate volumes across the three EE protocols (1–2 mL) were on average lower than the ~5–10 mL volumes reported earlier [14,15,47] but within the range of others [22,23,48]. These differences may be due to probe placement and whether the nerves involved in ejaculation were being stimulated in the same way. There also were differences across those studies in how the 80 stimuli were administered; our protocol used a 2-s-on/2-s-off pattern, which might have resulted in less stimulation overall. Other factors, such as environment [49], genetics [13], management, and age [50], can also affect semen production in nondomestic cats. The tigers in this study were all of prime reproductive age and in good health, although there was no information on pedigree. Finally, the frequency of collection can affect semen volume, as a short interval between collections produced a higher volume in young bulls [51] and domestic cats [52]. In this study, a 1-month break between collections was used, so more frequent collections might impact semen parameters.

Sperm concentration averaged 84.93 ± 26.63 × 10^6^/mL, which was similar to other studies in tigers [15,25,48,52,53], as well as in leopards [23,54] and lions [23]. Similarly, average sperm motility (56.43 ± 7.15%) was similar to other reports in tigers [47,53,55], leopards [14,22,23,53,55], lions [23], and pumas [56]. Finally, tigers in this study exhibited similar values for sperm viability (tiger [47,53,55]; lion [18]), normal morphology (tiger [23,53]; leopard [23,54]; lion [23]), and abnormal morphology (tiger [14,23,53,55]; leopard [23,54]; lion [23]) compared to other studies.

Seminal pH values were considerably higher than in previous studies of tigers [23,47], which could have been due to urine contamination, a common problem associated with EE protocols. Urine increases the pH and osmolarity of ejaculates and decreases sperm motility [57]. The amount of urine contamination may vary across series, so checking the pH of each fraction and only pooling those that are not urine contaminated could improve overall semen quality. To address that problem, Lueders et al. [18] used urethral catheterization in African lions and found it resulted in less urine contamination and more highly concentrated semen samples with excellent motility. Therefore, urethral catheterization may be an alternative for tiger semen collection in the future. That technique relies on the use of alpha2-adrenergic receptor agonists, such as medetomidine or dexmedetomidine, and has been shown to allow successful recovery of substantial sperm numbers, although electroejaculation may be better for maximal sperm recovery [58].

Reactive oxygen species (ROS) are free radicals containing oxygen [59] that can cause sperm cell damage [60]. Oxidants interfere with normal sperm function via membrane lipid peroxidation, having a deleterious effect on the morphological status of sperm cells and, thereby, on male fertilization potential [61]. An important marker of lipid peroxidation is MDA, which to our knowledge has yet to be evaluated in felids. Semen characteristics and concentrations of MDA in tiger serum and seminal pellets did not differ among the three EE protocols; however, the percentage of primary abnormalities and concentrations of MDA in seminal plasma were significantly higher using the Low voltage EE protocol. In humans, seminal MDA concentrations were positively correlated with abnormal sperm morphology [40,61,62], a finding not observed in this study, although relationships approached significance between MDA and % normal morphology and % secondary abnormalities in the Low group, and for secondary abnormalities in the High voltage group. Thus, the Low voltage protocol might not have fully stimulated the nerves involved in ejaculation, resulting in incomplete or poor-quality ejaculates. Seminal plasma plays an important role in protecting sperm against oxidative stress [63], with low volumes having higher concentrations of MDA and potentially exposing spermatozoa to higher oxidative damage [37]. However, in this study, the correlation between ejaculate volume and seminal plasma MDA concentration was not significant, so more studies examining these relationships are needed.

Serum creatine kinase (CK) has been previously used to assess welfare in the context of assisted reproductive technologies. For example, EE has been shown to increase serum CK as a consequence of muscle damage in Pampas deer (*Ozotoceros bezoarticus*) [64] and Iberian ibex (*Capra pyrenaica*) [65]. To our knowledge, this is the first time that serum CK has been studied in Bengal tigers in relation to different EE voltages. Our results showed that serum CK concentrations increased during the High voltage group series, while no changes were observed in the Low and Medium voltage groups, suggesting muscle damage might occur at higher voltages and could have a negative effect on animal welfare.

Serum testosterone concentrations in tigers can vary considerably from 0.86 to 6.75 ng/mL [14,23], which was comparable to our range of 0.76–3.50 ng/mL. In Bengal tigers, fecal androgen metabolite concentrations are known to increase and play a role during the mating season [66]. In response to EE, serum concentrations of testosterone and cortisol before, during, and after EE were not significantly different across the three protocols. These results support previous studies in tigers [14,15], indicating that EE under anesthesia does not elicit significant adrenal responses or influence gonadal steroidogenic activity during collection. However, across protocols, mean testosterone concentration was lower in the Medium compared to the High voltage protocol, but only slightly. Overall cortisol concentrations were lower than previously reported in tigers under similar semen collection procedures [14], but that could be due to assay/antibody differences between the commercial RIA used by Wildt et al. [14] and the EIA used in this study. By contrast, both serum cortisol and testosterone concentrations were similar to a study in Siberian tigers [15] that utilized a chemiluminescence immunoassay.

While the semen characteristics and hormones measured in this study were minimally affected by the different voltage protocols, it is possible that electrical stimulation, especially at high voltages, causes other physiological damage. For example, EE has been associated with increases in rectal temperature, respiratory, and heart rates, and biochemical (hematocrit and hemoglobin concentrations) changes, as shown in goats (*Capra hircus*) [67,68] and rams [69]. Such data are lacking for tigers, so more work is needed to determine the effect of EE voltage protocols on physiological function, as well as documenting vocalizations, walking and recumbence patterns, and other behaviors indicative of pain and discomfort to assess the animal wellbeing after the procedure.

## 5. Conclusions

Voltages used in other studies have ranged from 2–8 V, within the range of what was used in this study. Overall, there were no improvements in ejaculate quality with higher voltages. Semen characteristics were minimally affected by the different EE voltages, with the exception of sperm quality, which was lower in the Low voltage group and related to higher seminal plasma MDA concentrations and possible oxidative effects. Peripheral concentrations of testosterone, cortisol, and MDA were similar across voltage groups; however, serum CK was significantly higher in the High voltage series, indicating possible muscle damage. In general, the Medium voltage protocol (3–6 V) appeared to be the best option for collecting good quality sperm in tigers, without causing muscle damage, and so it is recommended for EE used to routinely cryobank or collect samples for artificial insemination. The lower voltage of the Medium protocol potentially exerts fewer negative effects on welfare, although more work is needed to assess physiological biomarkers in addition to cortisol and CK.

## Figures and Tables

**Table 1 animals-13-01893-t001:** Mean (± SEM) semen quality parameters and MDA concentrations among the three electroejaculation protocols using Low, Medium, and High voltage series.

Parameters	Low	Medium	High
Volume (mL)	2.31 ± 0.43	2.06 ± 0.51	1.88 ± 0.54
Concentration (×10^6^/mL)	80.00 ± 31.22	49.75 ± 15.59	125.41± 33.06
Total sperm (×10^6^)	133.51± 73.65	125.02 ± 56.19	184.99 ± 53.04
Motility (%)	42.88 ± 9.61	62.93 ± 5.39	63.54 ± 6.43
Viability (%)	58.56 ± 5.42	71.94 ± 2.57	68.97 ± 3.53
pH	8.61 ± 0.09	8.77± 0.07	8.70 ± 0.09
Sperm morphology			
Normal (%)	62.11 ± 3.43	66.66 ± 7.40	79.47 ± 2.84
Primary abnormality (%)	28.36 ± 2.72 ^a^	16.68 ± 2.91 ^b^	13.57 ± 1.72 ^b^
Secondary abnormality (%)	9.49 ± 1.13	6.48 ± 0.78	8.52 ± 1.65
MDA			
Pellet (nmol × 10^6^ sperm)	1.99 ± 0.31	1.53 ± 0.24	1.59 ± 0.25
Seminal plasma (nmol/L)	2.30 ± 0.31 ^a^	1.27 ± 0.29 ^b^	0.71 ± 0.32 ^b^
Serum (nmol/L)	3.36 ± 0.14	2.99 ± 0.14	3.20 ± 0.14

^a,b^ Significant differences between means among the three electroejaculation groups (*p* < 0.05).

**Table 2 animals-13-01893-t002:** Mean (± SEM) testicular size in tigers measured before electroejaculation.

Group	Right Testicle				Left Testicle			
	Width (cm)	Length (cm)	Height (cm)	Volume (cm^3^)	Width (cm)	Length (cm)	Height (cm)	Volume (cm^3^)
Low	3.20 ± 0.08	4.03 ± 0.15	3.72 ± 0.21	34.02 ± 1.83	3.28 ± 0.12	4.18 ± 0.18	3.54 ± 0.22	34.39 ± 3.09
Medium	3.11 ± 0.11	4.25 ± 0.19	3.45 ± 0.17	32.93 ± 2.56	3.00 ± 0.08	4.36 ± 0.14	3.38 ± 0.15	31.38 ± 1.13
High	3.25 ± 0.12	4.33 ± 0.13	3.59 ± 0.12	35.88 ± 1.83	3.23 ± 0.09	4.40 ± 0.10	3.44 ± 0.13	34.71 ± 0.85

**Table 3 animals-13-01893-t003:** Mean (±SEM) serum testosterone, cortisol, MDA, and CK concentrations before, during, and after each EE protocol (Low, Medium, and High voltage series).

Parameters	Low	Medium	High
Serum testosterone (ng/mL)			
Before	0.89 ± 0.19	0.83 ± 0.20	1.19 ± 0.24
Series 1	0.99 ± 0.18	0.83 ± 0.18	1.22 ± 0.24
Series 2	1.02 ± 0.19	0.91 ± 0.19	1.34 ± 0.28
Series 3	0.86 ± 0.15	0.76 ± 0.20	1.62 ± 0.07
After	0.78 ± 0.40	0.82 ± 0.16	1.05 ± 0.18
Total	0.99 ± 0.17 ^ab^	0.88 ± 0.17 ^a^	1.17 ± 0.17 ^b^
Serum cortisol (ng/mL)			
Before	1.78 ± 0.09	1.86 ± 0.09	1.82 ± 0.09
Series 1	1.62 ± 0.11	1.45 ± 0.12	1.68 ± 0.12
Series 2	1.05 ± 0.12	1.49 ± 0.09	1.53 ± 0.09
Series 3	1.47 ± 0.08	1.62 ± 0.10	1.62 ± 0.07
After	1.54 ± 0.12	1.43 ± 0.08	1.53 ± 0.08
Total	1.59 ± 0.06	1.57 ± 0.06	1.64 ± 0.06
Serum malondialdehyde (µmol/L)			
Before	3.50 ± 0.37	3.33 ± 0.23	3.39 ± 0.29
Series 1	3.31 ± 0.26	3.08 ± 0.14	3.43 ± 0.26
Series 2	3.16 ± 0.30	3.12 ± 0.12	3.38 ± 0.35
Series 3	3.45 ± 0.31	2.52 ± 0.27	2.98 ± 0.44
After	3.39 ± 0.25	2.99 ± 0.28	2.84 ± 0.19
Total	3.36 ± 0.14	2.99 ± 0.14	3.20 ± 0.14
Serum CK (ng/mL)			
Before	92.33 ± 9.35	96.90 ± 7.89	71.92 ± 4.18 ^C^
Series 1	93.50 ± 7.27	104.91 ± 8.92	78.92 ± 5.36 ^C^
Series 2	105.00 ± 11.00	103.83 ± 10.10	106.78 ± 17.10 ^BC^
Series 3	111.09 ± 9.56	115.00 ± 12.6	109.85 ± 9.45 ^AB^
After	118.67 ± 10.1	127.85 ± 11.7	118.46 ± 11.10 ^A^
Total	104.00 ± 4.32	110.44 ± 4.72	97.33 ± 4.11

^a,b^ Mean values differed across voltage series (*p* < 0.05). ^A,B,C^ Mean values differed within voltage series (*p* < 0.05).

## Data Availability

The datasets generated for this study are available on request to the corresponding author.

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
