# Peer review of "Effect of Electroejaculation Protocols on Semen Quality and Concentrations of Testosterone, Cortisol, Malondialdehyde, and Creatine Kinase in Captive Bengal Tigers"

_animals, 2023, doi:10.3390/ani13121893_

Round 1
Reviewer 1 Report
This is an interesting and well-written manuscript and a pertinent topic given the increasing concern for wellbeing in animals. Table 4 is of questionable value, as there are no significant findings, no real justification for the comparisons, and it is not discussed significantly in the manuscript. Consider omitting or greatly revising to point out any associations, or lack thereof, that are important to the conclusions, or, at minimum, consistent . Table 5, similarly, presents a large amount of data with very inconsistent trends, leading the reader to believer there is little relevance. If this table is to remain it should be discussed or minimized to pertinent findings.
More detail would be appreciated in describing the reason for frequent serum collection. Cortisol is intuitive, but the reasons for serum testosterone and MDA measurement are not particularly clear. Are there other biomarkers of wellbeing that should have been considered? For instance, serum creatinine kinase as a measure of muscle damage would provide more information. Blood collection from the day after the procedure might have revealed more than frequent collection between series. In the end, the conclusion that Medium is best because it 'might' be better for the animals is unconvincing without measures of wellbeing, even description of pain or discomfort experienced by the animals after the procedure.
Minor text comments are as follows:
1. Line 26: please define MDA prior to using the acronym.
2. Line 38: ‘serum testosterone, and malondialdehyde (MDA) concentrations, and cortisol as a proxy for stress’ It’s unclear in this sentence whether MDA and cortisol were serum measures.
3. Line 60: ‘protecting and surviving wild counterparts’ did you mean to omit the ‘and’?
4. Line 218 “ MDA in neat serum and seminal plasma and sperm pellets diluted in 500 µl of PBS” Were all three diluted in PBS? Or just pellets?
Author Response
"Please see the attachment."

Reviewer 2 Report
Dear Dr. Jaruwan Khonmee,
The aim of this study is to investigate the effects of three EE protocols (Low, Medium, High) on semen quality, testicular size, serum testosterone, cortisol concentrations, and seminal plasma and sperm pellet MDA measured after each EE series.
This research is considered to be very important for ex situ conservation of the endangered Bengal tiger. Moreover, I think that the use of a large number of animals, 12 animals, is also highly valuable.
But I think that you should be necessary at some modifications to publish your article in this journal.
Overall comments
In recent years, urethral catheterization method has been used to collect feline semen (including wild Felid). It has been reported that this method imposes less stress on animals than the transrectal electrical stimulation method used in this study, and may be able to collect more sperm. In this study, semen collection was performed by transrectal electrical stimulation instead of that method. Please tell us why you used only this method. In addition, although it is mentioned a little in the Discussion part, the introduction part should also mention the semen collection method by urethral catheterization. Also, please explain why you chose this method.
In this study, you are examining the size of the testicles and comparing the difference in semen collection due to the difference in voltage. Does the size of the testicles change depending on the difference in voltage? Please tell me about the voltage applied to the animal and its effect on the size of the testes. Rather, isn't the size of the testes just an examination to confirm that there are no abnormalities in seasonal changes or changes in physical condition at that time?
If the purpose is to perform artificial insemination, the number of sperm that can be collected seems to be one of the most important items. In this study, the ejaculation volume and sperm concentration are mentioned, but shouldn't you actually consider the total sperm count in one semen collection? If high voltage can collect more sperm, it will be better for artificial insemination.
Some comments
L69-75  You should also mention semen collection by urinary catheterization here.
L130-131 What is acetate Ringer's solution administered for?
L131-132 Please describe the administration method. Intravenous administration? Intramuscular injection?
L138 Insert “through the scrotal wall”.
L161 Modify “µL” to “µl”. Please unify.
L181 Modify “min” to “minutes”. Please unify.
L187,189,197 and more Modify “h” to “hours”. Please unify.
L201 Modify “H2SO4” to “H2SO4”. Note the subscript.
L242 Does this mean that each of the 12 animals was tested once with a different voltage? Since there are individual differences in semen quality, etc., I think it would have been better to conduct one test for each animal and have the same number of three groups.
L309 Modify “106” to “106”. Note the superscript.
L337-338 In this study, I have some doubts about the reason why the Low voltage one increased the number of primary abnormal sperm. The abnormal sperm that has increased in this study is macrocephalic head, but I feel that such abnormal sperm is originally an abnormality that occurs in the process of spermatogenesis.
Table 1 Is the semen volume of the Low group correct? If this is true, it seems like too much (urine contamination?). Conflicts with L297-298 statement.
I think that the total sperm count from one semen collection should be described.
Table 3 Make a superscript “a”.
Author Response
"Please see the attachment."

Reviewer 3 Report
Dear Authors,
In my opinion, an interesting work, well written. The information it contains may be helpful in obtaining semen from Bengal tigers. The methodology was correctly presented. The only weakness of the work may be the assessment of sperm motility using the microscopic method. For a better assessment of sperm quality, it would be more advisable to use a computer method of assessing movement and movement parameters. Nevertheless, apart from these analyses, the authors performed a number of other analyses, which may also provide valuable information about the quality of the semen obtained and the impact of the applied electroejaculation procedures on the welfare of animals.
I noticed minor editorial errors in the manuscript, which I list below:
The ‘x’ sign should be replaced with the appropriate ‘×’ sign, e.g. line 116, 117, 140, 163, 170, 181
Convert ‘ml’ to ‘mL’ - e.g. line 165, check throughout the text and in table 1.
Correct the notation ‘ ÌŠC’ to ‘°C ’- lines: 180, 182, check throughout the text.
Replace ‘μl’ with ‘μL’ - lines: 186, 188, 198, 199, check throughout the text.
The authors sometimes write in the methodology of ‘min’ (line 188) and sometimes ‘minutes’ (line 199). It should be standardized throughout the text.
The authors state that they used ‘three voltages protocols (Low voltage protocol = 59, Medium voltage protocol = 60, High voltage protocol = 59)’. Please provide more details. What was the difference between protocol Low and High - lines: 258-259.
In the bibliography, please check items 35 (should be written ‘Aitken R.J.,………………… ’), 47 (should be written ‘Wayan Kurniani Karja N. (?)………………’)
Author Response
"Please see the attachment."

Round 2
Reviewer 2 Report
Dear Dr. Jaruwan Khonmee,
Most of the parts that I've pointed out are revised properly.
I understand the author's thoughts on urethral catheterization method. Since it was added to the Discussion part, I think that this is good.
I think the addition of creatine kinase to the data in this paper was very good.